# Anesthesiological Preoperative Interview with a Palliative Care Patient: A Simulation-Based Experiment Using Standardized Patients

**DOI:** 10.3390/medicina60101577

**Published:** 2024-09-26

**Authors:** Christoph L. Lassen, Fabian Jaschinsky, Elena Stamouli, Nicole Lindenberg, Christoph H. R. Wiese

**Affiliations:** 1Department of Anesthesiology, University Medical Center of Regensburg, 93053 Regensburg, Germany; nicole.lindenberg@ukr.de; 2Department of Anesthesiology, Klinik Kitzinger Land, 97318 Kitzingen, Germany; fabian.jaschinsky@me.com; 3Department of Educational Science/Pedagogy II, University of Regensburg, 93053 Regensburg, Germany; eleni.stamouli@paedagogik.uni-regensburg.de; 4Department of Anesthesiology and Intensive Care, Herzogin Elisabeth Hospital, 38124 Braunschweig, Germany; c.wiese@heh-bs.de

**Keywords:** perioperative, advanced care planning, DNR-order, palliative medicine, standardized patients

## Abstract

*Background and Objectives*: Anesthesiologists come into contact with patients under palliative care in different clinical settings. They also routinely encounter these patients in their primary field of work, the operating room. Patients receiving palliative care who are scheduled for surgery will pose unique challenges in perioperative management, often presenting with advanced disease and with different psychosocial and ethical issues. This study aims to evaluate whether anesthesiologists without specialty training in palliative medicine will spot perioperative challenges presented by patients under palliative care and address them adequately. *Materials and Methods:* In this study, we simulated a preoperative anesthesiological interview using standardized patients and anesthesiologists (specialists as well as trainees). The standardized patients were asked to represent a patient under palliative care in need of surgery because of a mechanical ileus. We conducted 32 interviews, dividing the anesthesiologists into two groups. In one group, the standardized patients were instructed to address four problems, i.e., use of a port catheter for anesthesia, nausea and vomiting, pain medication, and an advance directive including a limitation of treatment (DNR-order). In the other group, these problems were also present, but were not actively addressed by the standardized patients if not asked for. The interviews were recorded, transcribed, and then analyzed. *Results:* In most cases, the medical problems were spontaneously identified and discussed. In only a few cases, however, was a therapy recommendation made for improved symptom control. The advance directive was spontaneously discussed by only 3 of the 32 (9%) anesthesiologists. In another 16 cases, the advance directive was discussed at the request of the standardized patients. The limitation of treatment stayed in place in all cases, and the discussion of the advance directives remained short, with an average duration of just over 5 min. *Conclusions:* In this study, the complex problems of patients under palliative care are not sufficiently taken into account in a preoperative anesthesiological interview. To improve treatment of the medical problems, therapists who have palliative medicine expertise, should be involved in the perioperative medical care, ideally as a multi-professional team. The discussion about perioperative limitations of treatment should be held beforehand, for example, as part of a structured advanced care planning discussion.

## 1. Introduction

Anesthesiologists will come into contact with patients under palliative care in different clinical settings, e.g., intensive care [1] and emergency medicine [2]. They also routinely encounter these patients in their primary field of work, the operating room [3,4]. Usually, the first contact of an anesthesiologist with patients undergoing surgical procedures is the preoperative anesthesiological interview. This interview includes a medical evaluation and obtaining informed consent for the necessary anesthesiological procedures. Patients receiving palliative care who are scheduled for surgery will pose unique challenges in perioperative management, often presenting with advanced disease and with different psychosocial and ethical issues, e.g., end-of-life-discussions. In order to provide optimum care, these issues need to be addressed and taken into account besides the usual anesthesiological care. Based on the involvement in pain medicine and symptom control in general, some anesthesiologists view their specialty as the ideal basis for providing palliative care [5,6]. Other authors have called for anesthesiologists to take a more active, even a leading, role in the perioperative care of patients under palliative care [7]. The aim of the present single-center observational study is to evaluate whether anesthesiologists without specialty training in palliative medicine will identify the perioperative challenges and issues (difficult vascular access, nausea and vomiting, pain, and limitation of treatment) presented by patients under palliative care and address them adequately. We chose to use a qualitative approach to accomplish this aim because we believed it would provide us with a vivid and in-depth view of the role of anesthesiologists involved in perioperative care of patients under palliative care.

## 2. Materials and Methods

In this study, we simulated a preoperative anesthesiological interview using standardized patients who were previously trained and deployed by the medical student training and education center of our university, as well as anesthesiologists. The thirty-two participants were either anesthesiologists in training or fully trained anesthesiologists. They were randomly selected from all anesthesiologists of the Department of Anesthesiology of our university hospital. None of the participants had prior training in palliative care. Physicians involved in the planning of this study were not eligible to participate. We formed two groups, randomly allocating sixteen anesthesiologists to each group (Group 1 and Group 2). In both groups, the standardized patients were advised to represent a patient under palliative care, whose characteristics are detailed in Table 1.

The anesthesiologists were instructed to conduct a regular preoperative anesthesiological interview, including documentation on our institution’s anesthesia chart and obtaining informed consent. Before the interview, the anesthesiologists were able to review certain patient documents (Table 2) without any time constraints.

The anesthesiologists did not receive any other information about the patient. The interview was then conducted as deemed appropriate by the anesthesiologist. In group 1, the interview was finished according to the anesthesiologist’s choice. In group 2, the standardized patient was instructed to ask questions on four specific issues at the end of the interview if they had not been previously addressed by the anesthesiologist. These questions were (1) “What should I do with my fentanyl patch?”, (2) “Is there a particular risk because of my constant vomiting?”, (3) “I have such bad veins. Can you use my port for the anesthesia?”, (4) “I have an advanced directive. It is important for me that I do not want to be resuscitated. Is this a problem?”. We chose these four topics because they are representative of medical and psychosocial issues in patients under palliative care and they are also common issues in anesthesiology (pain, vascular access, fasting). We feel that the inclusion of more issues (e.g., dyspnea, anxiety, depressive symptoms) would have made the study too complex. The manuscript is based on the Standards for Reporting Qualitative Research [8].

We collected demographic data of the participating anesthesiologists. All interviews were recorded on an audio and video recording system. All audio data were transcribed according to current transcription standards [9] using the f5 transcription program (dr. dresing & pehl, GmbH, Marburg, Germany), including automated timing of the interview. Data were then analyzed focusing on the aforementioned four topics (fentanyl patch, nausea and vomiting, use of port catheter for anesthesia, advance directive). For all topics, we analyzed whether it was discussed spontaneously by the anesthesiologist or only after the reminder by the standardized patients (group 2), as well as the solution offered and whether the topic was documented on the anesthesia chart. Statistical analysis was performed using SPSS Statistics 26 (IBM, Armonk, NY, USA), with *p* values < 0.05 considered significant. We used the chi-square test and two-sided t-test to compare the group compositions.

## 3. Results

All 32 interviews were analyzed. The mean work experience of the participating anesthesiologists was 7.5 years (SD 6.3), with 21 (66%) male and 11 (34%) female participants. Group 1 and Group 2 did not differ significantly in their composition concerning the anesthesiologists’ sex and work experience, with 12 male participants in group 1 and 9 male participants in group 2 (*p* = 0.264). The mean work experience was 6.1 years in group 1 and 8.9 years in group 2 (*p* = 0.226). The mean duration of all interviews was 21:38 min (SD 7:06 min). The duration of the interviews differed between group 1 (20:00) and group 2 (23:18) non-significantly (*p* = 0.098). The analysis of the four topics of interest (use of port catheter, nausea and vomiting, pain control, handling of advance directive) is shown in Table 3.

Most anesthesiologists considered the use of the port catheter possible, with the majority (53%) rejecting primary use. Reasons for this included concerns about damaging or clogging the catheter. Nausea and vomiting were noticed by all anesthesiologists spontaneously. Adjustments to the anesthesiological procedure were mentioned by most but not all anesthesiologists (80%), i.e., rapid sequence induction. Additional measures and therapies mentioned for nausea and vomiting noted on the anesthesiological chart were placement of a nasogastric tube (5 cases) and antiemetic medication (ondansetrone and melperone, 1 case each). Except for two anesthesiologists, most anesthesiologists mentioned the fentanyl patch spontaneously. When discussed, the fentanyl patch was left in place, with 25% of the anesthesiologists recommending additional therapeutic measurements for pain perioperatively (mostly regional anesthesia, i.e., epidural analgesia). None of the participants recommended removal of the patch. The existence of an advance directive was detected spontaneously by four anesthesiologists (13%). Because, in group 2, the existence of an advance directive was addressed directly by the patients, the advance directive was discussed in a total of 20 cases (63%). Out of these 20 cases, the DNR order included in the advance directive was upheld unchanged in 16 cases. Three anesthesiologists opted to resuscitate for a very limited time if the reason for cardiac arrest could be potentially reversed quickly, and one anesthesiologist opted to uphold the DNR order except for instances of anesthesiological or surgical iatrogenic incidents. The discussion of the advance directive took as a mean of 5:17 min (SD 3:46 min). With almost all anesthesiologists addressing the topics of port catheter, nausea and vomiting, and pain control, we only analyzed whether the work experience of the anesthesiologists addressing the advance directive spontaneously in group 1 differed from that of the ones not addressing the advance directive. The work experience of those addressing the advance directive was 5 years vs. 6.5 years of those not addressing the advance directive (*p* = 0.612).

## 4. Discussion

While anesthesiologists are often a part of a palliative care team, patients under palliative care are also encountered in the actual working environment of anesthesiologists, the operating room [3,4]. As suggested by other authors [7], this study investigated whether anesthesiologists without specialty training in palliative medicine would identify the perioperative challenges and issues presented by patients under palliative care and address them adequately.

In our study, medical problems could almost always be identified, even if not every problem solution appeared to be ideal. Contrary to what some anesthesiologists mentioned, a port catheter can be used for anesthesia, but knowledge of the handling process is necessary to prevent complications [10]. However, a port catheter is not suitable for administering higher volumes [11]. It seems worthwhile to integrate the basics of handling a port catheter into anesthesiological training. An ileus is a common clinical situation. It is surprising that not every anesthesiologist at least discussed a rapid sequence induction in the present case. Treatment suggestions for the nausea were only rarely given, but a nasogastric tube could be indicated for symptom control in addition to risk reduction and aspiration when inducing anesthesia [12].

The fentanyl patch was left in place in all patients, a practice which is in line with long-standing practice recommendations [13], but is debated internationally [14]. Undebatably, 250 µg/h of fentanyl represents a high-dose therapy, so it would be expected that perioperative pain therapy would become more difficult. Epidural analgesia could be a useful addition for this problem [15], as suggested by some anesthesiologists. Additionally, in the case of complex pain problems, consulting a pain management specialist before the operation is recommended [16].

We see the handling of the advanced directive and the limitation of treatment as a central point of this study. In Germany, there are no recommendations by medical societies for the perioperative handling of advance directives, as there are in other countries (e.g., UK, USA [17,18]). The fundamental unifying element of these recommendations is the discussion with the patient and, if necessary, their relatives in order to determine the patients’ wishes and goals and to create a joint, binding agreement on how to deal with the limitation of treatment perioperatively. The automatic suspension of limitation of treatment, as advocated in the past [19,20], is certainly no longer up to date and is not recommended [21,22]. In our study, only a few anesthesiologists spontaneously noticed the advance directive, although its existence was clearly recognizable by the information on the anesthesia questionnaire. This is similar to another simulation-based study that found that only 17 out of 30 anesthesiologists discussed an advance directive, which was contained in the patients’ files [23]. In this study, the advance directive was discussed for an average of just over 5 min, which seems very short and not sufficient. At least the advance directive was not suspended in any case. The transfer of information on the anesthesia protocol was also inadequate. The presence of the advance directive was documented, but not necessarily the content of the discussion. This means that it cannot be guaranteed that what was discussed would actually be implemented in the operating room. An example of this is shown by the following quote from an interview: Patient: “Well, can I rely on it being carried out that way [upholding the limitation of treatment]?”, Anesthetist: “Usually yes.”, Patient: “Usually?”, Anesthetist: “Usually yes.”.

Based on this short excerpt of the conversation, the problem becomes clear that a guarantee of the actions of the physicians working in the operating room can hardly be given. Therefore, exact documentation of the content of the conversation would be even more important.

Although there is commonly discussion of limitations of treatment in preoperative interviews, even on the day of surgery [24], we do not consider the anesthesiological preoperative interview to be the appropriate time to address the complex problems of patients under palliative care. The ethical challenges were especially not adequately addressed by the anesthesiologists in our study. The focus seemed to be on passing on information and obtaining consent and not on therapy. Anesthesiologists without additional training cannot, per se, be regarded as palliative medicine experts, which is not surprising, since palliative medicine requires more knowledge than one specialty can cover alone. Based on our results, we feel that the anesthesiological preoperative interview cannot provide the right framework for the complete care of a patient under palliative care, especially for an end-of-life discussion. Prerequisites for these discussions are defined in the literature, including how to train for these discussions [25,26,27]. These requirements cannot be met within this framework.

We see some limitations in our study. The study was only carried out at one location in Germany. It remains unclear whether the results at other hospitals or in other countries would have been different, especially in comparison to countries in which there are recommendations for the perioperative handling of advance directives. Also, different countries have different legal and cultural aspects concerning advance directives. Although many western countries (i.e., Europe/North America) have similar legal principles [28,29,30], other countries differ significantly, e.g., China [31], India [32], and African countries [33]. In Africa especially, the concept of advance directives is considered to be too individualistic [33].

We also did not obtain any feedback from the anesthesiologists in the sense of a debriefing. This was purposeful to prevent the handling of advance directives from being identified as one of the central issues of this study and being passed on among the participants before their respective participation.

## 5. Conclusions

In summary, we therefore recommend identifying patients under palliative care at an early stage. The complex problems of these patients were not sufficiently taken into account in anesthesiological preoperative interviews. It would be reasonable to include the perioperative handling of an advance directive and limitations of treatment in the process of advanced care planning. The exact handling of the limitation of treatment could be determined in advance and would then only have to be applied perioperatively. To improve treatment of the medical problems, physicians who have palliative medicine expertise, ideally anesthesiologists with additional training in palliative medicine, should be involved in the perioperative medical care.

## Figures and Tables

**Table 1 medicina-60-01577-t001:** Role-description for the standardized patients.

Subject Areas	Description
Age	64 years
Current medical history	First diagnosis of a rectum carcinoma six months ago. At the time of diagnosis, already metastasized disease with liver and lung metastasis. Initiation of palliative chemotherapy to inhibit tumor growth. After initially well-controlled symptoms, rapid clinical detoriation and increasing symptom load over the last two weeks with increasing pain levels and levels of nausea/vomiting and fatigue. After initial treatment of these symptoms and, again, improved symptom control, development of a high obstructive ileus. Subsequently, the patient is scheduled for surgical treatment with a palliative gastrostoma.
Previous illnesses	None
Social background	Married for 40 years, no children
Family medical history	Father died from hepatic cellular carcinoma, mother died from breast carcinoma. Both parents had been cared for by the patient until their deaths. Both parents had suffered for a long time due to their illnesses. Based on that experience, the patient has set up an advanced directive.
Character	Patient is calm and composed. Has come to terms with the disease. Is glad that the pain is well adjusted. Knows about the limited life expectancy. Hopes to obtain symptom relief from nausea and vomiting through the planned surgery.

**Table 2 medicina-60-01577-t002:** Patient documents.

Document Type	Summarized Content
Discharge letter from previous hospital	List of previous diagnoses including the presence of a pectoral port catheter. Outline of the previous medical history and treatment. Different examinations with no abnormal findings on ECG, echocardiography, or chest x-ray. CT-scan of the abdomen noticeable for tumor mass leading to obstruction of the small intestine and ileus, additional lesions in the lumbar spine, highly suspicious of bone metastasis.
Lab results	Noticeable for increased creatinine levels (1.3 mg/dL) and reduced hemoglobin levels (10.9 g/dL)
Patient chart	Current vital signs, current medication (Fentanyl transdermal patch 250 µg/h, Ibuprofen 800mg TID, Metoclopramide 10 mg TID, Macrogol TID, Pantoprazole 40 mg QD)
Informed Consent Form	This questionnaire, pre-filled by the patient, provides information, among other things, about previous illnesses and medication. The last question is: “Do you have an advanced directive?” This is answered with a yes. The form is also used for the documentation of the written informed consent.
Advanced Directive	The advance directive was not part of the documents automatically given to the anesthesiologist, but it was provided by the patient only upon request from the anesthesiologist. The form used for this advance directive is the official template for advance directives published by the German Federal Justice Department. The patient expressly rejects intensive therapy and treatment of cardiac arrest by chest compression and defibrillation under all circumstances. Mechanical ventilation and medical support of circulation are allowed only for the time of anesthesia/surgery. The reason for this rejection is mentioned in the advance directive (parents’ suffering).

**Table 3 medicina-60-01577-t003:** Analysis of the participants’ problem-solving of four specific topics during the interview ^1^.

	Group 1 (N = 16)	Group 2 (N = 16)	Total (N = 32)
Port Catheter			
Port catheter mentioned spontaneously	15 (94%)	16 (100%)	31 (97%)
Port catheter can be used for induction			
- Yes	2 (13%)	3 (19%)	5 (16%)
- Yes, but only if peripheral i.v. access is not possible	6 (38%)	11 (69%)	17 (53%)
- No	0 (0%)	2 (13%)	2 (6%)
- Use not discussed	8 (50%)	0 (0%)	8 (25%)
Port catheter’s use documented on anesthesia chart	0	1 (6%)	1 (3%)
Preoperative Nausea and Vomiting			
Nausea/vomiting mentioned spontaneously	16 (100%)	16 (100%)	32 (100%)
RSI mentioned spontaeneously	10 (63%)	13 (81%)	23 (72%)
RSI mentioned after direct question ^2^	N/A	2 (13%)	2 (6%)
Therapy for nausea/vomiting discussed ^3^			
- Nasogastric tube placement preoperatively	2 (13%)	1 (6%)	3 (9%)
- Medication	3 (19%)	2 (13%)	5 (16%)
- None	11 (69%)	13 (81%)	24 (75%)
Therapy for nausea/vomiting documented on anesthesia chart	4 (25%)	3 (19%)	7 (22%)
Transdermal Fentanyl Patch			
Fentanyl patch mentioned spontaneously	15 (94%)	15 (94%)	30 (94%)
Fentanyl patch mentioned after direct question ^2^	N/A	1 (6%)	1 (3)
Handling of the fentanyl patch			
- Keep patch attached preoperatively	5 (31%)	16 (100%)	21 (66%)
- Remove preoperatively	0 (0%)	0 (0%)	0 (0%)
- Not discussed	11(69%)	0 (0%)	11 (34%)
Additional pain therapy discussed			
- Epidural analgesia	4 (25%)	3 (19%)	7 (22%)
- Consult pain service preoperatively	0 (0%)	1 (6%)	1 (3%)
- Not discussed	12 (75%)	12 (75%)	24 (75%)
Handling of the fentanyl patch documented on anesthesia chart	8 (50%)	11 (69%)	19 (59%)
Advance Directive (AD) with DNR-Order			
AD mentioned spontaneously	4 (25%)	0 (0%)	4 (13%)
AD mentioned only after direct question ^2^	N/A	16 (100%)	16 (50%)
Intraoperative handling of the DNR order			
- Uphold DNR order	3 (19%)	13 (81%)	16 (50%)
- Uphold DNR order with adjustments	1 (6%)	3 (19%)	4 (13%)
- AD not mentioned	12 (75%)	0 (0%)	12 (38%)
AD’s existence documented on anesthesia chart	4 (25%)	16 (100%)	20 (63%)
AD’s handling documented on anesthesia chart	2 (13%)	12 (75%)	14 (44%)

^1^ All data are presented as numbers (%). ^2^ Only applicable for Group 2. ^3^ One anesthesiologist recommended a nasogastric tube and additional antiemetic medication. RSI = rapid sequence induction, N/A = not applicable, AD = advance directive, DNR = do not resuscitate.

## Data Availability

The raw data supporting the conclusions of this article will be made available by the authors upon request.

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
