# Peer review of "Anesthesiological Preoperative Interview with a Palliative Care Patient: A Simulation-Based Experiment Using Standardized Patients"

_medicina, 2024, doi:10.3390/medicina60101577_

Round 1
Reviewer 1 Report
Comments and Suggestions for Authors
Interesting study that is well written. Though the applicability and importance of the study might not be as high as some other studies. The methodology is sound and highlights an important point regarding anesthetic care of palliative patients. The findings do seem to support the conclusion and highlights the importance of clarifying specific details for care regarding palliative patients before they are in the peri-operative encounter.
Author Response
Comments: Interesting study that is well written. Though the applicability and importance of the study might not be as high as some other studies. The methodology is sound and highlights an important point regarding anesthetic care of palliative patients. The findings do seem to support the conclusion and highlights the importance of clarifying specific details for care regarding palliative patients before they are in the peri-operative encounter.
Response: Thank you very much for reviewing the manuscript. We appreciate your positive assessment of our study very much.
Reviewer 2 Report
Comments and Suggestions for Authors
I thank the editor for the oportunity to review this article Please consider the following issues:
Introduction: Since DNR-order is very heterogenous worldwide with regard to the specific law complexity in various societies, please provide the legal background (differences, similarities among various (significant) countries) for a better understanding of the readers. (i.e. there are countries were DNR is not legally mentioned, so no anesthesiologist will ask about it)
Methodology: What about palliative patients with increased anxiety? They were not discussed in your settings. Please be aware that with regard to various countries/ level of development/ religious background etc, the level of anxiety (preoperatively) could be very high. Please justify why did you chose this prototype of patient only (calm, etc- page 3)
Results: Although very interesting, please consider to conduct a comparative analysis, using various appropriate tests
Discussion: please reconsider to shape your discussion in the light of the results (comparative analysis), aiming to highlight the importance of specific training for the anesthesiologist in the palliative care medicine.
Author Response
Thank you very much for reviewing our manuscript. We appreciate the given suggestions to improve our manuscript.
Comment 1: Introduction: Since DNR-order is very heterogenous worldwide with regard to the specific law complexity in various societies, please provide the legal background (differences, similarities among various (significant) countries) for a better understanding of the readers. (i.e. there are countries were DNR is not legally mentioned, so no anesthesiologist will ask about it)
Response 1: Thank you very much for pointing out this important aspect of different approaches to end-of-life care in different cultures or countries. We address the aspect, that our study was conducted in Germany in the limitations section. Therefore we felt, that this aspect is best placed in this section as well. Please let us know if you feel this is inadequate (lines 222-227).
Comment 2: Methodology: What about palliative patients with increased anxiety? They were not discussed in your settings. Please be aware that with regard to various countries/ level of development/ religious background etc, the level of anxiety (preoperatively) could be very high. Please justify why did you chose this prototype of patient only (calm, etc- page 3)
Response 2: We limited the number of issues to avoid making the case too complex for the anesthesiologists studied. We included anxiety as a possible issue in the Materials and Methods section and explained the limited number of issues.
Comment 3: Results: Although very interesting, please consider to conduct a comparative analysis, using various appropriate tests
Response 3: I am very sorry but I am not quite sure that I understand this comment correctly. We performed a comparative analysis for the group composition. We did not compare the results shown in table 3 for the respective groups. Given the similarity for the medical issues and the obvious differences in the results for the advance directive disscussion we felt that performing a statistical analysis would not add value to the data. Please let us know which specific comparison you feel should be performed.
Comment 4: Discussion: please reconsider to shape your discussion in the light of the results (comparative analysis), aiming to highlight the importance of specific training for the anesthesiologist in the palliative care medicine.
Response 4: Also see Response 3. We highlighted the importance of additional training be adding a sentence in the conclusion (line 241).
Reviewer 3 Report
Comments and Suggestions for Authors
Thank you for the opportunity to read and review the paper titled “Anesthesiological Preoperative Interview with a Palliative Care Patient: A Simulation-Based Experiment Using Standardized Patients” by Lassen and co-authors. The topic is highly relevant and has the potential to significantly impact the quality of life of palliative care patients and their families.
Unfortunately, the PDF version available for review contained missing words and lines in several sections. I hope to have the opportunity to review a revised version of the manuscript. Below are my further comments:
Introduction
If there are other concerns or issues that anesthetists could address during the preoperative assessment, which are not explored in this study, they should be mentioned in the introduction. Additionally, the rationale for focusing on specific issues should be explained. This section should also address the reasoning behind the choice of the clinical scenario and provide a detailed description of the aims of the paper. E.g. “ … the present single-center observational study aims to identify if (description of participants) will identify and address (topics 1,2,3)… in the setting of (description of setting and scenario).
Materials and Methods
Sections of the manuscript on page 3, lines 79-84, are missing, as well as Table 2 and lines 86-97.
Results
Again, sections of the results are missing, including lines 113-116.
The term "N/A" (not applicable) can be added to areas of Table 3 to prevent confusion.
Discussion
The stated aim of the study in line 157 should be consistent with the aim presented in the introduction.
Lines 182 and 189 are missing.
Lines 202-212
In this section, the authors should clearly distinguish between information drawn from the literature and conclusions derived from the present study.
Conclusions
Future directions can also be discussed in this section.
Author Response
Thank you very much for reviewing our manuscript. It is very unfortunate that you did not receive a complete version of our manuscript.
Comment 1:
Introduction
If there are other concerns or issues that anesthetists could address during the preoperative assessment, which are not explored in this study, they should be mentioned in the introduction. Additionally, the rationale for focusing on specific issues should be explained. This section should also address the reasoning behind the choice of the clinical scenario and provide a detailed description of the aims of the paper. E.g. “ … the present single-center observational study aims to identify if (description of participants) will identify and address (topics 1,2,3)… in the setting of (description of setting and scenario).
Response 1:
We changed the part of the introduction concerning the aims of the study according to the suggestions. We explained the reasoning behind the choice of the clinical scenario in the Materials and Methods in lines 94-96, which were missing in this reviewers version. We added other concerns in this part (lines 95-99).
Comment 2:
Materials and Methods
Sections of the manuscript on page 3, lines 79-84, are missing, as well as Table 2 and lines 86-97.
Response 2:
We hope you will be provided with a complete manuscript.
Comment 3:
Results
Again, sections of the results are missing, including lines 113-116.
The term "N/A" (not applicable) can be added to areas of Table 3 to prevent confusion.
Response 3:
We added N/A where appropriate
Comment 4:
Discussion
The stated aim of the study in line 157 should be consistent with the aim presented in the introduction.
Response 4:
Thank you very much for pointing out this inconsistency. We changed this part (lines 160-163).
Comment 5:
Lines 202-212
In this section, the authors should clearly distinguish between information drawn from the literature and conclusions derived from the present study.
Response 5:
We clarified which information is based on the results of the study (lines 209, 214)
Round 2
Reviewer 2 Report
Comments and Suggestions for Authors
Thank you very much for the opportunity to review the second form of the manuscript. Since no consistent changes were made comparing with the first form of this manuscript, it is still very difficult to be followed (especially discussion section), and the interest for Medicina readers is not very well represented due the heterogenity of ethical/legal issues concerning end-life care support and management, I would leave the decision at the editors discretion. Moreover, the descriptive way of presenting a limited number of interviews of patients/subjects, which are not representative for the large mass of palliative patients (even for Germany not)- considering the psychological aspects, reduces significantly the value of the content.